# Multivariate genome-wide association study of leaf shape in a *Populus deltoides* and *P. simonii* F$_1$ pedigree

**Wenguo Yang[1,2], Dan Yao[1], Hainan Wu[1], Wei Zhao[1], Yuhua Chen[1], Chunfa Tong[1]***

**1** Co-Innovation Center for Sustainable Forestry in South China, College of Forestry, Nanjing Forestry University, Nanjing, Jiangsu Province, China, **2** School of Artificial Intelligence and Information Technology, Nanjing University of Chinese Medicine, Nanjing, Jiangsu Province, China

* tongchf@njfu.edu.cn

**Data Availability Statement:** The RADseq data is available in the SRA database at NCBI (http://www.ncbi.nlm.nih.gov/Traces/sra) with the accession numbers listed in S1 Table. Other relevant data are

## Abstract

Leaf morphology exhibits tremendous diversity between and within species, and is likely related to adaptation to environmental factors. Most poplar species are of great economic and ecological values and their leaf morphology can be a good predictor for wood productivity and environment adaptation. It is important to understand the genetic mechanism behind variation in leaf shape. Although some initial efforts have been made to identify quantitative trait loci (QTLs) for poplar leaf traits, more effort needs to be expended to unravel the polygenic architecture of the complex traits of leaf shape. Here, we performed a genome-wide association analysis (GWAS) of poplar leaf shape traits in a randomized complete block design with clones from F$_1$ hybrids of *Populus deltoides* and *Populus simonii*. A total of 35 SNPs were identified as significantly associated with the multiple traits of a moderate number of regular polar radii between the leaf centroid and its edge points, which could represent the leaf shape, based on a multivariate linear mixed model. In contrast, the univariate linear mixed model was applied as single leaf traits for GWAS, leading to genomic inflation; thus, no significant SNPs were detected for leaf length, measures of leaf width, leaf area, or the ratio of leaf length to leaf width under genomic control. Investigation of the candidate genes showed that most flanking regions of the significant leaf shape-associated SNPs harbored genes that were related to leaf growth and development and to the regulation of leaf morphology. The combined use of the traditional experimental design and the multivariate linear mixed model could greatly improve the power in GWAS because the multiple trait data from a large number of individuals with replicates of clones were incorporated into the statistical model. The results of this study will enhance the understanding of the genetic mechanism of leaf shape variation in *Populus*. In addition, a moderate number of regular leaf polar radii can largely represent the leaf shape and can be used for GWAS of such a complicated trait in *Populus*, instead of the higher-dimensional regular radius data that were previously considered to well represent leaf shape.

within the paper and its Supporting Information files.

**Funding:** Funding for this research was provided by the National Natural Science Foundation of China (No. 31870654 and 31270706) and the Priority Academic Program Development of the Jiangsu Higher Education Institutions (PAPD).The funders had no role in study design, data collection and analysis, decision to publish, or preparation of the manuscript.

**Competing interests:** The authors have declared that no competing interests exist.

## Introduction

Leaves are the most fundamental photosynthetic organs in plants; they are responsible for absorbing solar energy to generate power for plant growth and thus provide food for many species on earth [1, 2]. Leaf morphology exhibits tremendous diversity between or within species, such as the broad leaves of poplars and needle leaves of conifers. Leaf size and shape are evolutionarily adapted to environmental changes in response to water and light stress [3, 4], making it possible to reconstruct the paleoclimate [5, 6]. In model systems, several genes and networks have been identified to affect initial leaf development and pattern formation [2, 7, 8] as well as leaf length and width [9–11] using the mutagenesis screening method. Moreover, quantitative trait loci have also been detected for leaf morphological traits in species such as tomato [12], *Arabidopsis* [13], *Brassica* [14], maize [15], barley [16], and *Populus* [17, 18]. Despite advances made in these studies, the identified genes or loci may only cover a portion of the leaf morphological variation observed in nature because the variation is considered to be under polygenic control [11, 19].

The genus *Populus* (2n = 38) is an ecologically and economically important tree with a wide distribution in diverse environments of the Northern Hemisphere [20, 21]. The genus, comprising approximately 30 species, was grouped into six sections (i.e., Abaso, Aigeiros, Leucoides, Populus, Tacamahaca, and Turanga) according to morphological parameters [22]. Most species have several attractive biological characteristics, such as fast growth and easy asexual reproduction, so they are of particular interest to forest breeders for developing new cultivars to meet the needs of pulp, paper, lumber, and biofuels industries. Several studies have shown that leaf traits are highly related to growth and habitat and can be used as predictors of productivity and determinant factors in phylogenetic relationships [11, 23, 24]. Therefore, persistent efforts have been made to dissect the genetic mechanism of morphological traits in the genus. In the 1990s, Wu *et al*. [25] first conducted QTL mapping of leaf morphology in $F_2$ hybrids of *P. trichocarpa* × *P. deltoides*, with up to 3 QTLs identified for each trait, leaf area and the ratio of leaf length to width, at four crown positions. Recently, Mckown *et al*. [26] found 6 and 5 SNPs significantly associated with leaf length and width, respectively, in a GWAS on unrelated wild accessions of *Populus trichocarpa*. Drost *et al*. [11] detected 2 QTLs for lamina length, 2 for width, and 5 for their ratio in a pseudobackcross population of *P. trichocarpa* and *P. deltoides*. More recently, Chhetri *et al*. [17, 18] performed a GWAS of many traits with different genotypes from natural populations of *P. trichocarpa*; they did not detect any significant SNPs for single leaf traits, including leaf length, width, perimeter, area, and aspect ratio, but the detected up to 9 SNPs for leaf morphology multitraits. In contrast, Fu *et al*. [27] precisely described leaf shape with radii from the centroid to the contour at regular intervals and performed a marker-trait association analysis of principal components of the high-dimensional radius data, leading to several QTLs identified for leaf shape in a natural population of *P. szechuanica* var *tibetica*. They further modeled the leaf contour of a QTL genotype as a dynamic trajectory and identified a few significant QTLs for the variation of leaf shape in the same population [28]. These studies could be considered an initial stage for unravelling the genetic mechanism behind leaf size and shape in *Populus*. More powerful strategies, such as the utilization of novel statistical methods and generation of more accurate phenotype and genotype data, should be taken into account to ensure the accuracy and precision of such a tough task.

Herein, we report a genome-wide association study (GWAS) of leaf size and shape with a randomized complete block design (RCBD), which was established using clones from an $F_1$ hybrid population of *P. deltoides* and *P. simonii* [29] belonging to the sections Aigeiros and Tacamahaca, respectively. The leaves of the female *P. deltoides* are broad, while those of the

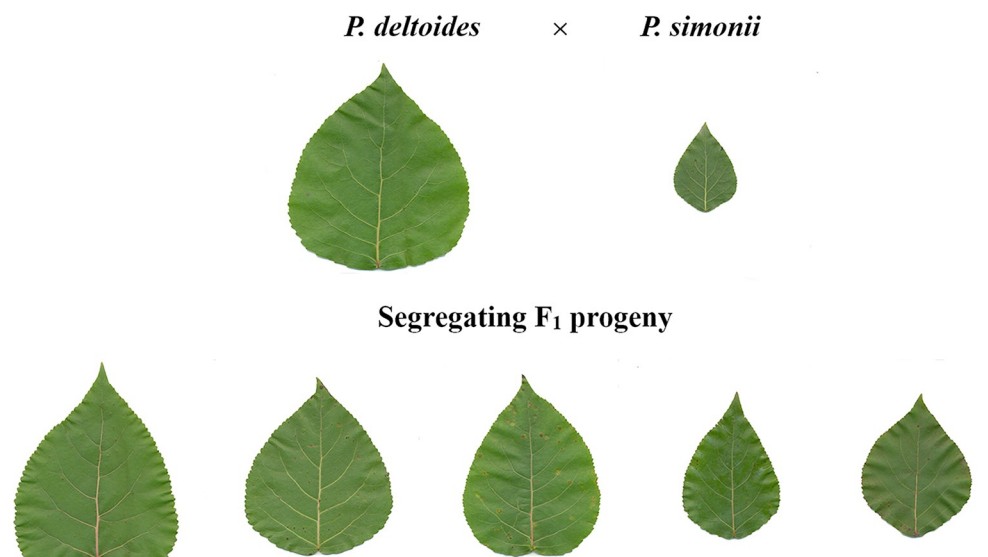

**Fig 1. Leaf shape variation among parents and progeny of *P. deltoides* × *P. simonii*.**

male *P. simonii* parent are narrow. This sharp contrast led to diverse leaf shapes in their progeny (Fig 1). The leaf traits were digitally derived from scanned images of leaves, including the classical indices of leaf length, width, and area, as well as the high-dimensional data of regular ordered radii between the leaf centroid and edge points, as described in Fu *et al*. [27]. Single nucleotide polymorphism (SNP) genotypes of each clone were generated by mapping the paired-end (PE) reads from restriction site-associated DNA sequencing (RADseq) to the reference genome of *P. trichocarpa* [21]. With these SNP data, we applied a linear mixed model (LMM) to conduct GWAS for multiple or single leaf traits using the R package EMMREML (https://cran.rproject.org/web/packages/EMMREML). Consequently, we identified many more SNPs significantly associated with leaf shape than those detected in previous studies. Furthermore, candidate genes associated with these SNPs were investigated to show that most flanking regions of these significant SNPs harbored genes that were related to leaf growth and development and to the regulation of leaf morphology. The results enhanced our understanding of the genetic mechanism of leaf shape variation in *Populus*. We demonstrated that the combined use of the traditional experimental design and the multivariate linear mixed model (mvLMM) could greatly improve the power of GWAS for leaf shape. Additionally, the multivariate data of a moderate number of regular leaf polar radii can largely represent the leaf shape and can be used for GWAS of such a complicated trait in *Populus*. This is contrary to the expectation that the high-dimensional regular radius data could well represent the leaf shape for GWAS, as indicated by Fu *et al*. [27].

## Materials and methods

### Genetic experimental design and measurement of leaf traits

To obtain repeated phenotypic data for accurate QTL analysis, we established an RCBD in the spring of 2017 with clones from an F₁ hybrid population of the female *P. deltoides* and the male *P. simonii*, which was produced in the spring of 2009 to 2011 [29, 30]. The design consisted of 234 clones with 3 blocks, 6 cuttings for each clone per plot within a block, and a 50 × 60 cm spacing in Xiashu Forest Farm of Nanjing Forestry University, Jurong County,

Jiangsu Province, China (32.1224˚N, 119.2155˚E). The sixth most apical mature leaf of each individual was sampled in mid-July 2017 by placing it into a paper bag and then scanned using a Hewlett-Packard Scanjet G2410 A4 flatbed scanner at a resolution of 300 dpi. The A4-sized images were saved as bmp files, and leaf size and shape traits were analyzed with the R package LeafShape (https://github.com/tongchf/LeafShape). These traits included leaf area (A), length (L), maximum width (W), and widths at one-third length (W1/3), half length (W1/2), and two-thirds length (W2/3) from the base, as well as 360 regular polar radii (RD360) between the leaf centroid and edge points, as shown in Fig 2. As our primary analytical step, we applied multivariate statistical methods to these leaf traits with SAS 9.3 software (SAS Institute, Cary, USA), including canonical correlation analysis and principal component analysis.

## SNP genotyping

Since 2013, more than four hundred individuals in the poplar hybrid population have been sequenced with RADseq technology in several batches [29, 30]. The 163 clones from the RCBD experiment and their two parents were sequenced previously, and their RADseq paired-end (PE) reads were deposited in the NCBI SRA database with the accession numbers listed in S1 Table. The PE reads for each individual were filtered using the NGS QC toolkit with default parameters [31], and the resulting high-quality (HQ) read data were used for calling SNP genotypes. The SNP calling procedures were largely the same as those described in Mousavi et al. [30]. In brief, the PE reads of each clone or parent were first aligned to the reference sequence of *P. trichocarpa* with the software BWA v0.7.17 [32]. Second, the resulting SAM (sequence alignment/map) file was converted into a BAM (binary alignment/map) formatted file and then sorted and indexed with SAMtools v1.9 [33]. Third, the sorted BAM file was analyzed to generate a VCF file using BCFtools v1.9 software [33]. Finally, the VCF file was filtered to obtain SNP genotypes of each individual such that a heterozygous allele had a read coverage depth (DP) of at least 3 and the quality of each SNP genotype was greater than 30.

Because the 163 clones were from an $F_1$ hybrid population, the SNPs were expected to segregate in several different patterns, such as *aa*×*ab* and *ab*×*ab*, due to the characteristics of outbred forest species [34, 35]. We classified the SNPs into subsets according to the segregation patterns and kept those that did not seriously deviate from Mendelian segregation ratios ($p > 0.01$), including 1:1, 1:2:1, and 1:1:1:1. In addition, if an SNP had more than 10% missing genotypes across the clones, it was removed from the dataset. To obtain independent SNP markers and linkage disequilibrium (LD) blocks, we performed the LD-based SNP pruning procedure for all the SNPs using PLINK v1.07 software with a window size of 25 SNPs, a step size of 2 SNPs, and an $r^2$ threshold of 0.7 [36].

## Statistical methods for GWAS

Since the poplar leaf is generally symmetrical, the polar radii on the right side largely contain the full information of all the radius data on both sides (Fig 2D). We used the multiple traits of different numbers of regular radii across $-\pi/2$ to $\pi/2$ to find SNPs associated with leaf shape, which was implemented with the mvLMM as follows:

$$y_{ijkl} = \mu_l + B_{il} + M_{jl} + G_{jl} + e_{ijkl} \tag{1}$$

where $y_{ijkl}$ is the $l$th polar radius of the $k$th tree leaf of the $j$th clone in the $i$th block; $\mu_l$ is the overall mean of the $l$th polar radius; $B_{il}$ is the effect of the $i$th block; $M_{jl}$ is the genotype effect of the $j$th clone at any tested SNP; $G_{jl}$ is the polygenic background effect of the $j$th clone; and $e_{ijkl}$ is the residual effect. It is assumed that $B_{il}$ and $M_{jl}$ are fixed effects, while $G_{jl}$ and $e_{ijkl}$ are

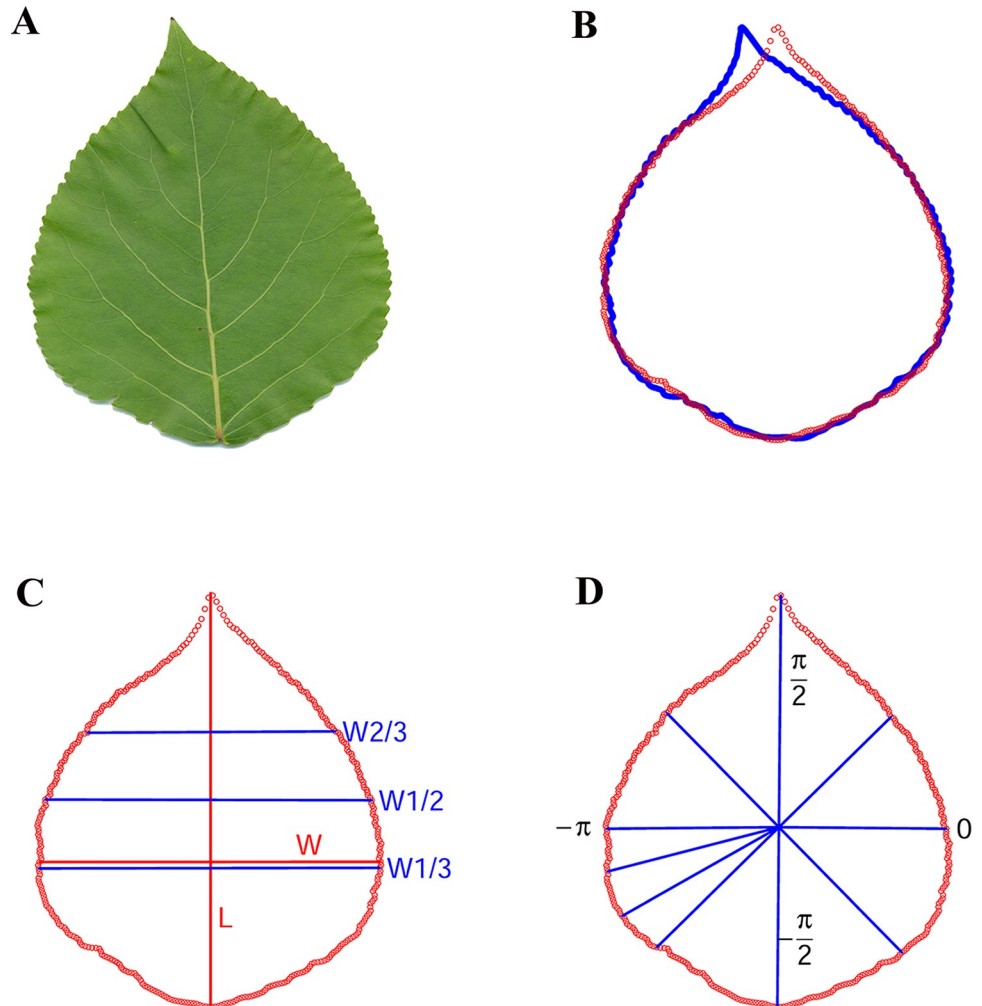

**Fig 2.** Workflow of the R package LeafShape for extracting leaf shape traits in poplar hybrids: (A) a fresh leaf; (B) original (blue) and position-adjusted (red) edge points extracted from the scanned image of a leaf; (C) the red vertical line indicates the leaf length (L), the red horizontal line indicates the maximum leaf width (W), and the blue horizontal lines represent the leaf widths at one-third length (W1/3), half length (W1/2), and two-thirds length (W2/3); (D) 360 regular polar radii between the leaf centroid and edge points across $-\pi$ to $\pi$.

random effects with $G_{jl} \sim N(0, \sigma_{g_l}^2)$, $e_{ijkl} \sim N(0, \sigma_{e_l}^2)$ and cov($G_{jl}, e_{ijkl}$) = 0. In matrix form, model (1) can be written as

$$Y = XB + ZG + E \qquad (2)$$

where $Y$ is the $n{\times}t$ matrix whose $(I, j)$th element is the $j$th trait value of the $i$th individual, i.e., the $i$th row of $Y$ is the multiple trait data for the $i$th individual; $X$ is an $n{\times}p$ known design matrix of fixed effects, including the overall mean, block effects, and individual genotype effects at the tested SNP site; $B$ is the $p{\times}t$ matrix of fixed effect coefficients; $G$ is the $c{\times}t$ matrix whose $(i, j)$th element is the background random additive genetic effect of the $i$th clone and the $j$th trait with Vec($G$)~$N_{c{\times}t}(0, V_G{\otimes}A)$, where Vec denotes the matrix vectorization function [37], $A$ is the additive relationship matrix for the $c$ clones and $V_G$ is the additive genetic variance matrix for the $t$ traits; $Z$ is the $n{\times}c$ coefficient matrix corresponding to the matrix $G$; $E$ is the random residual matrix with Vec($E$)~$N_{n{\times}t}(0, V_E{\otimes}I_n)$. Based on the assumptions above, the

covariance matrix of $\text{Vec}(Y)$ can be derived as

$$V = \text{cov}(\text{Vec}(Y)) = V_G \otimes ZAZ' + V_E \otimes I_n \tag{3}$$

Because the clones in our experiment belonged to a full-sib family and their parents were unrelated and not inbred, the kinship coefficient of any two clones is 0.25 in theory [38], leading to the relationship matrix $A$ with ones on the diagonal and 0.5 elsewhere.

To test whether any single SNP was significantly associated with the leaf traits, an $F$ statistic was used under the null hypothesis $M\text{Vec}(B) = 0$ for a full-rank $q \times pt$ matrix $M$, as

$$F = \frac{1}{q}(M\text{Vec}(B))'[M((I_t \otimes X)V^{-1}(I_t \otimes X'))^{-1}M']^{-1}(M\text{Vec}(B)) \tag{4}$$

with $q$ numerator degrees of freedom and $t(n-p)$ denominator degrees of freedom [39]. The $p$-values from the $F$ statistics (4) for each SNP are prone to genomic inflation [40, 41]. It is necessary to calculate the genomic inflation factor ($\lambda_{GC}$) to evaluate the inflation level. When there was no genomic inflation, the $p$-value threshold for testing significant SNPs was determined based on Bonferroni correction at the 0.05 significance level.

The proportion of phenotype variance explained (PVE) by a single SNP was calculated as

$$R^2 = 1 - \frac{RSS}{RSS_0} \tag{5}$$

where $RSS_0$ and $RSS$ are the residual sums of squares under the null hypothesis model and the full model (2), respectively [42].

As a comparison with the regular radius data, we also used mvLMM (2) to perform GWAS for the multiple traits of L, W, W1/3, W1/2, and W2/3 (LWs) (Fig 2C). For a single leaf trait such as length, width, and area, the GWAS was conducted with univariate LMM, which can be derived by simplifying the multivariate model (2) and is expressed as

$$y = X\beta + Zg + e \tag{6}$$

where $y$ is a vector of trait values for $n$ individuals; $X$ is a design matrix of fixed effects; $\beta$ is a vector of fixed effects; $g$ is a vector of random genetic effects for each clone with $g \sim N(0, \sigma_g^2 I_c)$; $Z$ is the coefficient matrix corresponding to the random vector $g$; $e$ is the random residual vector with $e \sim N(0, \sigma_e^2 I_n)$. Moreover, we calculated the narrow heritability of a single trait as $h^2 = \sigma_g^2/(\sigma_g^2 + \sigma_e^2)$ without incorporating the fixed effects of SNPs in model (6).

To calculate the restricted maximum likelihood (REML) estimates of genetic parameters, we applied the function *emmremlMultivariate* for the multivariate model (2) and *emmreml* for the univariate model (6) in the R package EMMREML (https://cran.r-project.org/web/packages/EMMREML). After the genetic parameters were calculated, the $p$-value for testing each SNP was calculated according to the $F$ statistic, as in Eq (4).

## Investigation of candidate genes

In our previous study [43], the average LD block length was estimated to be ~650 bp in the same hybrid population, which is so short that it could not be properly used as a downstream or upstream range for investigating candidate genes for the significant SNPs. Alternatively, we took the strategy as described in Slaten et al [44]. In brief, we considered candidate genes that contain significant SNPs or are within a LD block harboring significant SNPs. If a significant SNP is within an intergenic region and does not form a LD block, both the closest downstream and upstream genes are considered as candidates. Because no annotation information for leaf shape is available in the gene annotation database of *P. trichocarpa* in Phytozome (v4.1;

https://genome.jgi.doe.gov), the coding sequences (CDSs) of these genes were obtained for further annotating. We first performed BLAST searches with their CDSs against the nonredundant protein database [45, 46] and then mapped all BLAST hits to Gene Ontology (GO) terms based on ID mapping information from http://ftp.pir.georgetown.edu/databases/idmapping/idmapping.tb.gz. The descriptions of the blast hits and GO terms were saved in an Excel file in which we could search which genes were possibly related to leaf shape.

## Results

### Leaf trait data

We successfully obtained the leaf trait data for a total of 2,244 individual trees belonging to 163 clones in the RCBD (S2 Table). Some plots had missed samples due to the damage from pest, disease, poor rooting ability, or other unknown reasons. To validate these measurements, we measured 100 randomly chosen leaves with ImageJ [47] and LeafShape software separately. The average relative differences in the leaf length, width, and area values measured from the two software programs were 1.45 (±0.99)%, 4.76 (±0.76)%, and 5.05 (±0.92)%, respectively, indicating that the two measurements from both methods were largely consistent (S3 Table). The descriptive statistics for the traits L, W, W1/3, W1/2, W2/3, A, and the L/W ratio are presented in Table 1, including the mean, standard deviation, range, and coefficient of variation (CV). The CVs for the leaf length and different leaf widths were similar, ranging from 20.79% for L to 25.14% for W2/3, while the CV for leaf area reached a maximum value of 42.25% and the CV for the length/width ratio had a minimum value of 10.12%. The histograms showed that these leaf traits basically followed a normal distribution (S1 Fig). Furthermore, the heritabilities of leaf length and different leaf widths as well as leaf area were similar (40~50%), but the heritability of the length/width ratio was much higher at 64.74% (Table 1). In addition, correlation analysis showed that the leaf length, measures of leaf width, and leaf area were significantly positively correlated ($p < 0.01$) with each other, with most coefficients over 0.90; the minimum coefficient value was 0.8137 between L and W2/3 (S4 Table). However, the L/W ratio was significantly negatively correlated with each of the leaf length, different leaf widths, and leaf area traits, with coefficients between -0.6160 and -0.2389. Finally, analysis of variance showed that the effects of each leaf trait, L, W, W1/3, W1/2, W2/3, and A, were significantly different among blocks and clones (S5 Table).

The 360 polar radii (Fig 2D) of all leaves were obtained with the R package LeafShape as a full dataset denoted as RD360. We also extracted 5 reduced datasets denoted as RD06, RD09, RD11, RD16, and RD61 that contained 6, 9, 11, 16, and 61 regular polar radii of each leaf on

**Table 1. Variation in leaf length, different leaf widths, leaf area, and the ratio of length/width in the F$_1$ progeny of *Populus deltoides* × *Populus simonii* based on a randomized complete block design.**

| Trait (Unit) | Mean | SD | Range | CV (%) | Heritability (%) |
|---|---|---|---|---|---|
| L (mm) | 114.27 | 23.76 | 36.31 ~ 193.12 | 20.79 | 41.56 |
| W (mm) | 90.79 | 21.90 | 23.72 ~ 163.03 | 24.12 | 46.46 |
| W1/3 (mm) | 89.95 | 21.89 | 20.49 ~ 157.89 | 24.34 | 45.65 |
| W1/2 (mm) | 83.80 | 19.83 | 23.72 ~ 147.72 | 23.66 | 46.67 |
| W2/3 (mm) | 64.97 | 16.33 | 16.76 ~ 131.03 | 25.14 | 49.53 |
| A (mm$^2$) | 7317.88 | 3091.52 | 588.66 ~ 23514.73 | 42.25 | 49.52 |
| L/W | 1.28 | 0.13 | 0.91 ~1.92 | 10.12 | 64.74 |

Notes: L, leaf length; W, maximum leaf width; W1/3, leaf width at one-third length; W1/2, leaf width at half length; W2/3, leaf width at two-thirds length; L/W: The ratio of the leaf length to the maximum leaf width.

the right side from $-\pi/2$ to $\pi/2$, respectively; these datasets were expected to represent the leaf shape characteristics despite the lower dimensionality of the data due to leaf symmetry. Canonical correlation analysis showed that each of the commonly measured leaf traits, such as length, width, and area, was highly correlated with the polar radii in the RD360, RD61, RD16, RD11, RD09, and RD06 datasets, with a correlation coefficient value of over 0.98 and a $p$-value less than 0.0001 (S6 Table). Additionally, the multiple traits of leaf length and different leaf widths (LWs: L, W, W1/3, W1/2, and W2/3) were extremely significantly correlated with the 6 radius datasets, with the first canonical correlation coefficient greater than 0.9996 ($p < 0.0001$). Moreover, the principal component analysis of the polar radius traits revealed that the proportion of total variance for the first principal component was at least 95.59% for the 6 radius datasets. The first principal component for each radius dataset was highly correlated with leaf length, different leaf widths, and leaf area, with coefficients greater than 0.90 and $p$-values less than 0.0001 (S7 Table).

## SNP genotype data

A total of 33,086 SNPs across the 163 clones were obtained by mapping their high-quality PE reads separately to the reference genome of *P. trichocarpa* (v4.1; https://genome.jgi.doe.gov). For a SNP genotype of each clone at each SNP site, the heterozygous allele was required to have a coverage depth of at least 3 reads, whereas the coverage depth for a homozygous allele was at least 5. Furthermore, the quality of each genotype needed a Phred score of at least 30, and the missing genotype rate at each SNP was set to less than 10%. All the SNPs were categorized into five segregation types, *aa×ab*, *aa×ac*, *ab×aa*, *ab×ab*, and *ab×cc* (Table 2). The majority of SNPs segregated at a ratio of 1:1 ($p > 0.01$) with *aa×ab* and *ab×aa types*.

The LD analysis of these SNPs was performed with the software PLINK [36], resulting in 10,735 independent SNP markers and LD blocks. Therefore, the $p$-value threshold for significant SNPs in our genome-wide analyses was set to 0.05/10735 = 4.66E-6 (-$\log_{10}$($p$-value) = 5.33) based on the Bonferroni correction at the 0.05 significance level.

## Significant SNPs associated with leaf traits

mvLMM (2) was applied to perform the GWAS for the multiple traits of the regular polar radius datasets RD06, RD09, RD11, RD16, and RD61 separately, as well as the multiple traits of LWs. The quantile-quantile plots of the $p$-values on base 10 logarithm scale showed that there existed different levels of genomic inflation, with inflation factors greater than 1 for datasets RD06, RD09, and LWs; less than 1 for datasets RD16 and RD61; and almost equal to 1 for dataset RD11 (Fig 3). Because the result from dataset RD11 showed good genomic control, we used this result to determine the significant SNPs associated with leaf shape. Consequently, a total of 35 SNPs were found to be significantly associated with the multiple traits of leaf shape under the $p$-value threshold of 4.66E-6, each explaining 0.18–0.32% of the phenotypic variance

**Table 2. Summary of SNPs obtained across the 163 clones based on a randomized complete block design.**

| Segregation type | Ratio | Number |
|---|---|---|
| *aa×ab* | 1:1 | 13,385 |
| *aa×bc* | 1:1 | 76 |
| *ab×aa* | 1:1 | 19,295 |
| *ab×cc* | 1:1 | 159 |
| *ab×ab* | 1:2:1 | 171 |
| Total | | 33,086 |

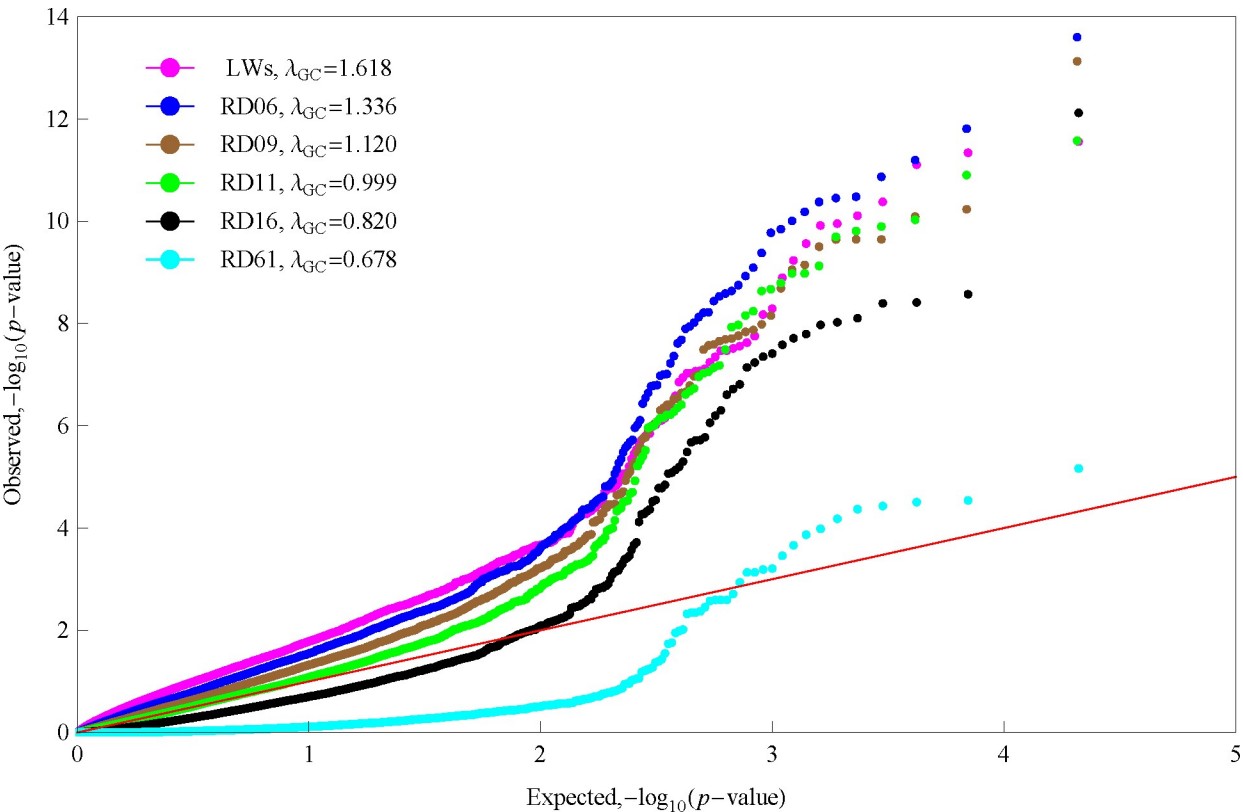

**Fig 3. Quantile-quantile plots of observed *p*-values versus expected *p*-values on a base 10 logarithm scale with genomic inflation factors for GWAS of the different multiple traits representing the poplar leaf shape.** LWs indicate the multiple traits of leaf length and 4 different leaf widths. RD06, RD09, RD11, RD16, and RD61 indicate the multiple traits of 6, 9, 11, 16, and 61 regular polar radii between the leaf centroid and edge points across $-\pi/2$ to $\pi/2$, respectively.

(Table 3). Fig A shows the Manhattan plot of the negative base 10 logarithm of the *p*-value against the corresponding SNP position. These significant SNPs were distributed on 8 chromosomes, with a few SNPs on chromosomes 4, 6, 10, 15, 16, and 18 but more on chromosome 14. There were 11 significant SNPs detected on chromosome 1, of which the first 10 were within a 3-Mb region. More surprisingly, chromosome 14 harbored the most (16) significant SNPs, which could be divided into five regions according to the position where the negative logarithm of the *p*-value changed from decreasing to increasing (Fig 4B).

Moreover, LMM (6) was applied to detect associations with each single leaf trait, such as leaf length, width and area. The results showed that the genomic inflation factors for these traits ranged from 1.774 for W2/3 to 2.461 for the L/W ratio (Fig 5). After genomic control, we found that there were no significant SNPs associated with any single trait under the *p*-value threshold based on Bonferroni correction (Fig 5). However, without genomic control, various numbers of significant SNPs were found for these traits: only one significant SNP each was detected for W, W1/3, and A; 6, 8, and 33 SNPs were detected for W1/2, W2/3, and the L/W ratio, respectively; and no significant SNPs were detected for L (S8 and S9 Tables; S2 Fig). The SNP at position 10669990 on chromosome 10 was a common SNP significantly associated with the four different leaf widths, and the significant SNPs for the width at two-thirds length shared all but one significant SNP with the width at half length. In addition, the most significant SNP sites or regions for the ratio of leaf length to leaf width were consistent with those for

**Table 3. Summary of the significant SNPs associated with the leaf shape represented by the multiple trait dataset RD11.**

| Chr | Position/Region | Number | P-Value | PVE (%) | Gene | Description[a] |
|---|---|---|---|---|---|---|
| 1 | 1561634 | 10 | 1.04E-6 | 0.20 | Potri.001G044300; | photosynthetic NDH subunit; response to light stimulus; response to light intensity; photosynthesis, light harvesting; regulation of auxin polar transport; MYB-like protein; transcription factor TCP20; photosynthetic NDH subunit; auxin-responsive protein; leaf senescence |
| | 2506253 | | 1.07E-8 | 0.24 | Potri.001G049900; | |
| | 2703246 | | 7.32E-8 | 0.22 | Potri.001G051300; | |
| | 2742828 | | 2.66E-12 | 0.32 | Potri.001G056700; | |
| | 3168327 | | 9.98E-7 | 0.20 | Potri.001G059100; | |
| | 3176353 | | 1.27E-10 | 0.28 | Potri.001G059800; | |
| | 3620732 | | 4.01E-6 | 0.18 | Potri.001G060000; | |
| | 3654397 | | 1.05E-9 | 0.26 | Potri.001G060200; | |
| | 4255086 | | 1.18E-8 | 0.24 | Potri.001G060400; | |
| | 4573250 | | 1.1E-7 | 0.22 | Potri.001G060900 | |
| 1 | 11059517 | 1 | 5.76E-9 | 0.25 | Potri.001G135925; | integral component of membrane; cellular response to phosphate starvation |
| | | | | | Potri.001G135950 | |
| 4 | 15591673 | 1 | 2.02E-10 | 0.28 | Potri.004G134200; | VQ motif-containing protein; response to water deprivation |
| | | | | | Potri.004G134300 | |
| 6 | 12471729 | 1 | 1.56E-10 | 0.28 | Potri.006G146400; | mRNA cleavage factor complex; mitogen-activated protein kinase |
| | | | | | Potri.006G146500 | |
| 6 | 25163980 | 1 | 1.05E-9 | 0.26 | Potri.006G253700; | SNARE-like superfamily protein; ethylene-responsive transcription factor |
| | | | | | Potri.006G253800 | |
| 10 | 4486477 | 1 | 6.14E-7 | 0.20 | Potri.010G030200; | cellular manganese ion homeostasis; integral component of membrane |
| | | | | | Potri.010G030400 | |
| 14 | 501461 | 4 | 5.3E-7 | 0.20 | Potri.014G000700; | MYB family transcription factor PHL6 isoform; transcription factor MYB44; L10-interacting MYB domain-containing protein; photosynthesis, light harvesting; protein weak chloroplast movement under blue light |
| | 988295 | | 8.81E-8 | 0.22 | Potri.014G022500; | |
| | 1554847 | | 1.61E-9 | 0.26 | Potri.014G026000; | |
| | 1943593 | | 6.99E-9 | 0.25 | Potri.014G029700; | |
| | | | | | Potri.014G029800; | |
| 14 | 2326098 | 5 | 9.41E-8 | 0.22 | Potri.014G034500; | regulation of leaf morphogenesis; transcription factor MYB44-like; response to auxin; MYB-related protein MYBAS1-like; L10-interacting MYB domain-containing protein; Cpn60_TCP1 domain-containing protein |
| | 2609112 | | 2.14E-9 | 0.26 | Potri.014G035100; | |
| | 3467134 | | 1.86E-7 | 0.21 | Potri.014G039800; | |
| | 3546254 | | 1.26E-11 | 0.30 | Potri.014G054700; | |
| | 3729803 | | 7.5E-10 | 0.27 | Potri.014G056100; | |
| | | | | | Potri.014G058500; | |
| 14 | 4014340 | 4 | 3.04E-6 | 0.19 | Potri.014G061450; | L10-interacting MYB domain-containing protein; response to red or far red light; regulation of leaf development; auxin-responsive protein; response to red or far red light; response to light stimulus; response to absence of light; protein spotted leaf 11-like |
| | 4309715 | | 9.39E-11 | 0.29 | Potri.014G066600; | |
| | 4316988 | | 7.08E-7 | 0.20 | Potri.014G066700; | |
| | 4564789 | | 2.45E-7 | 0.21 | Potri.014G066900; | |
| | | | | | Potri.014G067600; | |
| | | | | | Potri.014G073700; | |
| | | | | | Potri.014G075500; | |
| | | | | | Potri.014G076500 | |
| 14 | 5495011 | 1 | 2.32E-9 | 0.26 | Potri.014G081200; | MYB domain-containing protein; regulation of leaf morphogenesis; MYB family transcription factor |
| | | | | | Potri.014G087700; | |
| | | | | | Potri.014G089300 | |

*(Continued)*

**Table 3.** (Continued)

| Chr | Position/ Region | Number | P- Value | PVE (%) | Gene | Description[a] |
|---|---|---|---|---|---|---|
| 14 | 6606004 | 2 | 6.61E-8 | 0.22 | Potri.014G096300; | MYB-like transcription factor; auxin response factor; protein kinesin light chain-related 1 iosform; transcription factor MYB8-like; spotted leaf protein; auxin-responsive protein; leaf senescence; leaf development |
| | 7047979 | | 3.23E-8 | 0.23 | Potri.014G100100; | |
| | | | | | Potri.014G100400; | |
| | | | | | Potri.014G100800; | |
| | | | | | Potri.014G102000; | |
| | | | | | Potri.014G103300; | |
| | | | | | Potri.014G103500; | |
| | | | | | Potri.014G103900 | |
| 15 | 6867668 | 1 | 3.87E-7 | 0.21 | Potri.015G052600; | calcium ion binding; protein heterodimerization activity |
| | | | | | Potri.015G052800 | |
| 16 | 3663208 | 1 | 4.44E-7 | 0.21 | Potri.016G055200; | integral component of membrane; accelerated cell death 11 |
| | | | | | Potri.016G055300 | |
| 18 | 4076713 | 1 | 8.32E-7 | 0.20 | Potri.018G046800; | histidine-containing phosphotransfer protein; zinc finger family protein |
| | | | | | Potri.018G046900 | |
| 18 | 15471331 | 1 | 1.10E-6 | 0.20 | Potri.018G145568 | NBS-LRR type disease resistance protein |

[a] The descriptions were chosen from the annotations in S10 Table.

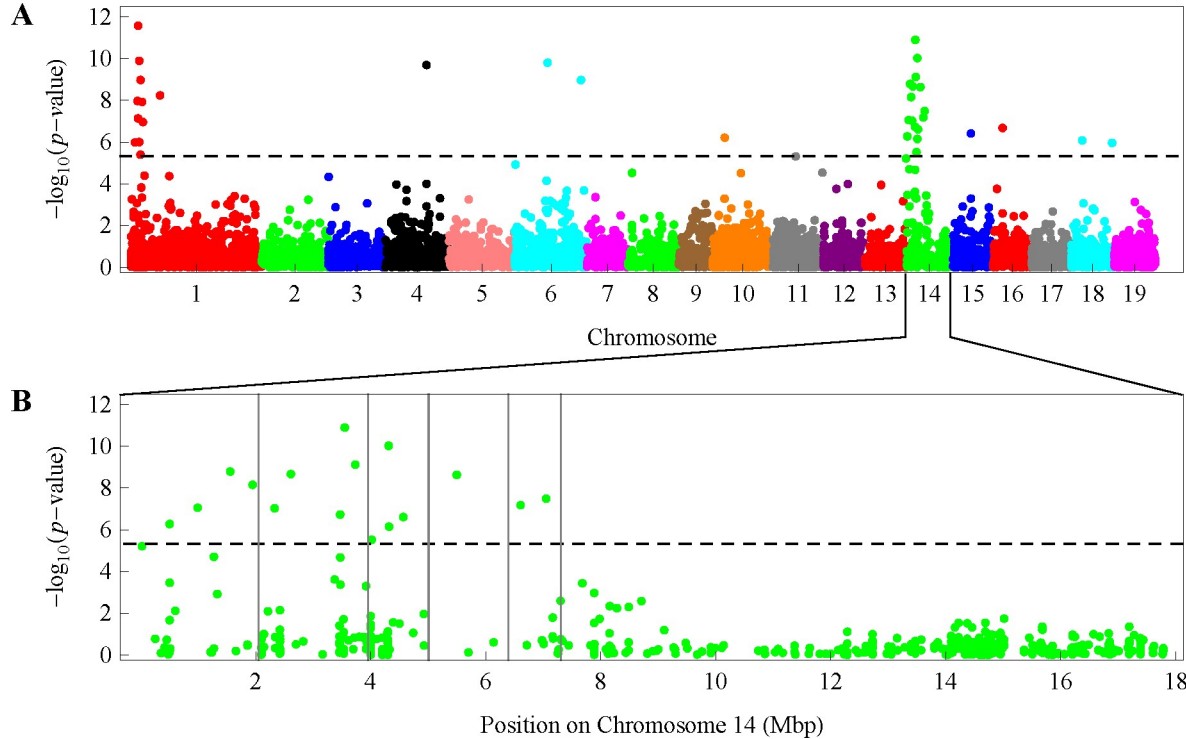

**Fig 4. Manhattan plot of the association analysis for the 11 regular polar radii between the leaf centroid and edge points from -π/2 to π/2.** (A) The plot shows the 19 chromosomes of the reference genome of *P. trichocarpa*. The horizontal dashed line indicates the genome-wide significance threshold of 5.33, which is a base 10 logarithm of the *p*-value based on the Bonferroni correction at the 0.05 significance level. (B) The significant SNPs on chromosome 14 were divided into five regions roughly according to the position where the negative logarithm of the *p*-value changes from decreasing to increasing.

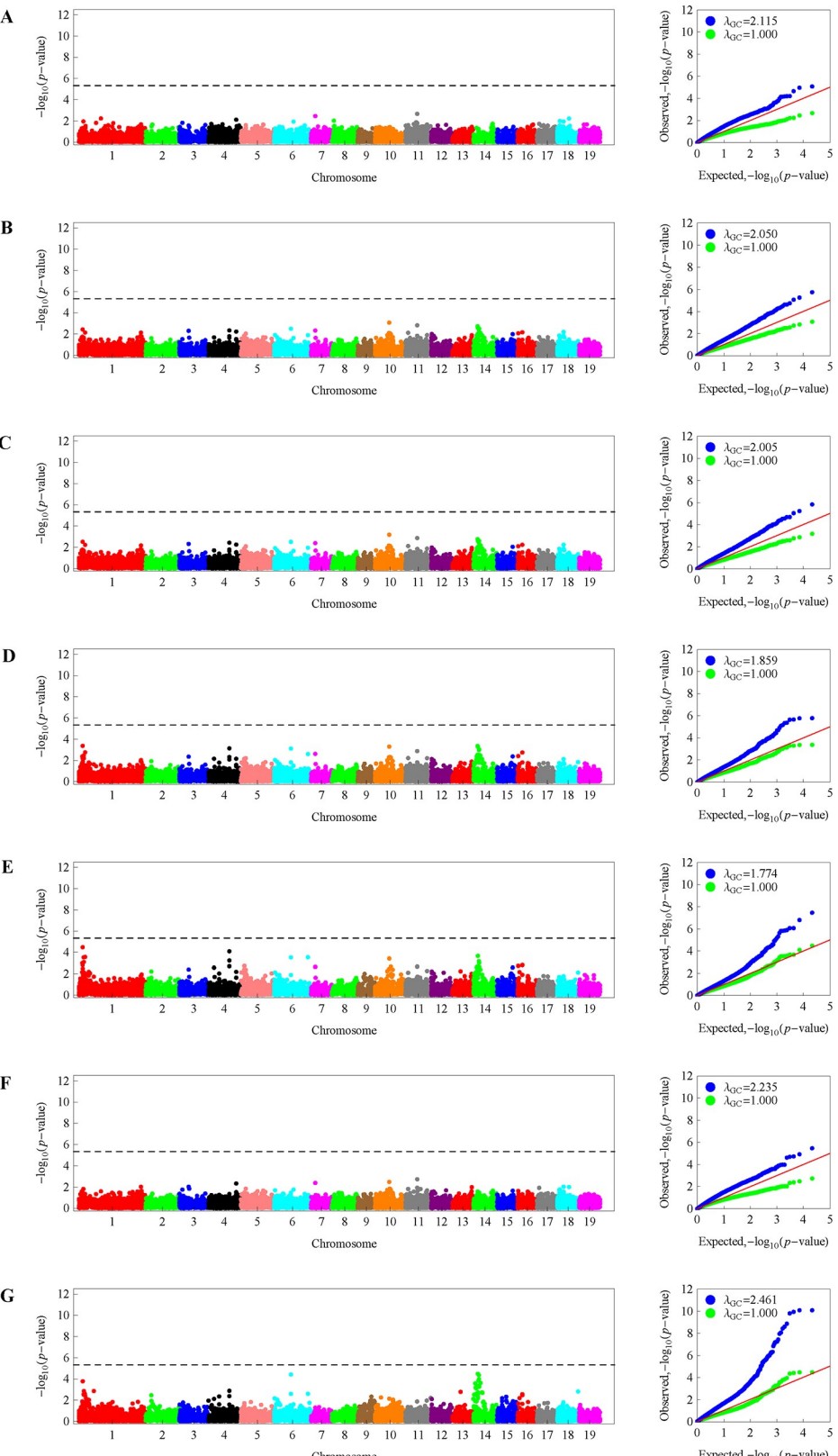

**Fig 5.** Manhattan and quantile-quantile (QQ) plots of the association analyses for each univariate trait, L (A), W (B), W31 (C), W21 (D), W32 (E), and A (F), and the ratio of L to W (G) across the 19 chromosomes of the reference

genome of *P. trichocarpa*. The left panel presents the Manhattan plots under genomic control, while the right panel shows the corresponding QQ plots before (blue) and after (green) genomic control. The horizontal dashed line indicates the genome-wide significance threshold of 5.33, which is a base 10 logarithm of the *p*-value based on the Bonferroni correction at the 0.05 significance level.

the multiple traits of leaf length and four different leaf widths, except for the two significant SNPs on chromosomes 2 and 13.

## Candidate genes affecting leaf shape

The candidate genes of the significant SNPs for the multiple traits of the 11-dimensional regular polar radii data were annotated with the nonredundant protein database at the NCBI and GO databases (S10 Table). One significant SNP region on chromosome 1 and five on chromosome 14 were found to harbor a total of 40 candidate genes functionally related to leaf shape (Table 3). However, the rest 9 significant SNPs on chromosome 1, 4, 6, 10, 15, 16, and 18 had no candidate genes that have descriptions directly related to leaf shape, possibly due to the reason that each of them did not form a LD block with other SNPs and thus had at most two candidate genes. We found that there are 8 candidate genes in 5 significant SNP regions, which directly affect leaf growth and development, with descriptions such as "leaf development" and "regulation of leaf morphogenesis". It was also noticed that there are 6 candidate genes on chromosomes 1 and 14 related to the hormone auxin, which plays important roles in initial leaf formation, lamina margin elaboration, and leaf vasculature patterning [48–51]. Moreover, 12 candidate genes were found to belong to MYB gene family, which was previously reported to be involved in leaf development in *Arabidopsis* [52] and maize [53]. Furthermore, 2 candidate genes on chromosomes 1 and 14 are related to TCP genes, which were found to be involved in leaf development and morphology in *Arabidopsis* [54, 55]. In addition, 14 candidate genes were related to light responses or photosynthesis in 5 significant SNP regions distributed on chromosomes 1 and14; these genes are involved in activities such as response to light intensity, light harvesting, and photosynthesis. Undoubtedly, these genes play important roles in leaf development and pattern formation.

## Discussion

Leaf size and shape are the most important traits during the development and growth of *Populus*. Understanding the genetic mechanism of these traits is of great interest to many poplar breeders. In the present study, we successfully detected dozens of SNPs significantly associated with the multiple traits of the 11-dimensional regular leaf polar radii in a randomized complete block test with clones from the $F_1$ hybrids of *P. deltoides* and *P. simonii*. Multiple traits could be considered to represent the leaf shape because the regular polar radii on the right side largely reflect the two-dimensional pattern of the leaf. Compared with previous studies for identifying QTLs or SNPs associated with leaf shape in *Populus* (see Introduction), we were able to identify many more QTLs or significant SNP regions. One of the main reasons for the powerful ability to identify the associated SNPs may be attributed to the use of the RCBD in the current GWAS. This kind of test design provided replicates of clones not only at the block level but also at the plot level, allowing thousands of individuals to be used for the association analysis. From a statistical perspective, the repeated phenotype data for each genotype that originated from a single seed can control for the spatial effects in the field and reduce systematic errors, hence improving the accuracy and power of GWAS. In contrast, in previous GWAS or QTL mapping studies on poplar leaf traits, phenotype data were measured from

single plants with different genotypes in natural populations or full-sib families, possibly limiting QTL detection power.

Another advantage of our association analysis strategy may be due to incorporating the multiple traits of leaf polar radii into the mvLMM for GWAS. Although mvLMMs have become increasingly important in GWAS because of their power gain over univariate analysis, the computation of genetic parameter estimates is nontrivial [56]. We successfully implemented the parameter and statistical calculations with the flexible R package EMMREML by adding or modifying some codes. Consequently, the mvLMM helped identify many more significant SNPs associated with leaf traits without genomic deflation. In contrast, after genomic control, the univariate LMM did not have the ability to detect any significant SNPs for any single trait, such as leaf length and width (Fig 5). Even if genomic deflation was permitted, we could see that fewer than 10 significant SNPs were detected for the single traits W, W1/3, W1/2, W2/3, and A, whereas no significant SNPs were detected for L (S8 Table). However, in such cases, the number of significant SNPs dramatically increased to 33 for the ratio of leaf length to width but was still less than the number detected based on the multiple traits of the 11-dimensional regular leaf polar radii dataset (S9 Table). The fact that more SNPs were detected for the ratio of leaf length to leaf width than for the other single traits may largely be due to much higher heritability of this trait (Table 1). This phenomenon can also be found in a previous study [11], where the authors identified 2 QTLs for leaf length and 2 for width but 5 for the ratio of the two traits.

Although our association analysis of the multiple traits based on the mvLMM was able to identify many more significant SNPs, it seems that the PVE of each SNP was much lower, ranging from 0.18 to 0.32% (Table 3). An intuitive explanation for this result is that leaf shape is possibly controlled by many genes with small effects, conforming to the infinitesimal model [57]. This explanation could further confirm that our strategy for GWAS in the current study is powerful for detecting such small-effect genes. This phenomenon may be the main reason why previous studies had a lower power for locating QTLs for single leaf traits, with only a few detected, although the PVEs of the QTLs were apparently larger than those estimated in this study [11, 25, 28]. However, the PVEs of SNPs or QTLs cannot be compared directly because they are calculated based on not only different population structures but also different statistical models. Even in the same study using the same statistical model, the PVE may or may not consistently increase or decrease with the corresponding statistical value for determining the significance of the hypothesis test. This is because the estimates of the environmental variance vary for different SNPs or QTLs, possibly leading to inconsistencies between the PVEs and statistical values. This phenomenon can be commonly found in the literature. For example, in Drost *et al.* [11], the first QTL for lamina length had a PVE value of 6.31% with a LOD value of 3.14, while the PVE of the second QTL was 8.10% with a lower LOD value of 2.68. In addition to these factors, the most important consideration is how to calculate the PVE based on a statistical model. For most fixed linear models with uncorrelated phenotype data, the $R^2$ statistic is generally used to measure the PVE in QTL mapping studies or GWAS. However, for mixed linear models, such measurements are not well established [58]. Here, we calculated the $R^2$ statistic as Eq (5) based on the weighted residual sum of squares [59].

It is worth emphasizing that the 11-dimensional multivariate data of the regular leaf polar radii can largely represent the poplar leaf shape and can be applied in association analyses with SNPs for such traits that are difficult to measure. Naturally, it was believed that the higher the dimensionality of the radius data between the leaf centroid and edge points is, the better the characteristics of leaf shape can be represented (Fig 2D). Fu *et al.* [27] first implemented such an idea by extracting 360 coordinates on leaf outlines from scanned images and performed a series of association analyses with the leaf shape [28, 60]. We also performed GWAS of leaf

shape with different dimensions of the radius data (e.g., RD61, RD16), which were extracted by our own R package (https://github.com/tongchf/LeafShape) because Fu *et al.* did not provide public software for the task. However, our results showed that for the higher dimensional data (i.e., RD61 and RD16), genomic deflation existed with $\lambda_{GC} \leq 0.820$, while for the lower dimensional data (i.e., RD09 and RD06), genomic inflation existed with $\lambda_{GC} \geq 1.120$ (Fig 3). In contrast, the RD11 data presented a balanced result between genomic inflation and deflation, exhibiting the best performance regarding genomic control in the GWAS with different dimensional data of the regular leaf polar radii.

Compared with previous studies for poplar leaf shape, we found that there were a few overlapping regions (<5 Mb) containing significant SNPs or QTLs. S11 Table lists those significant SNPs or QTLs associated with leaf shape in the current study and in four recent studies [17, 18, 26, 61], excluding those previous QTL studies in which no physical QTL position information was available [11, 25, 27, 28]. The results in the previous studies for single leaf traits such leaf length and width were not considered because we thought that the leaf shape could not be described by a single leaf parameter. We found that there were 7 significant SNPs detected in our study very close (<5 Mb) to one or more SNPs identified in previous studies, of which 5 were consistent with Xia et al. [61], 4 with Chhetri et al. [17], 1 with McKown et al. [26], and 1 with Chhetri et al. [18]. In contrast, 5 overlapping regions were found between the four previous studies. It is interesting to find that 3 regions on chromosome 4, 6, and 8 were coincidentally detected for leaf shape in three studies. Although our GWAS findings have more consistent SNPs with the previous results, most SNPs identified in the current and previous studies did not share an overlapping region. This result may be due to many reasons, but one of the main reasons is that different methods were used to describe the complex trait of leaf shape in the GWAS or QTL studies. Drost et al. [11] described the leaf shape with the ratio of leaf length to width, while Chhetri et al. [17, 18] described it with the combination of leaf area (LA), leaf dry weight (LD), leaf length (LL) and leaf width (LW) or the combination of leaf aspect ratio (AR) and specific leaf area (SL). However, based on the method of Fu et al. [27, 28], we used high-dimensional regular polar radii data to describe the leaf shape.

## Conclusion

The novel strategy for GWAS with direct integration of the traditional randomized complete block design and the multiple traits of regular leaf polar radii into the multivariate linear mixed model facilitated the identification of many more significant SNPs associated with leaf shape in *Populus* than previous studies have detected. Moreover, it was demonstrated that the multivariate linear mixed model was more powerful than the univariate linear mixed model in the association analyses for leaf traits such as leaf length, width, and area. Most flanking regions surrounding significant SNPs harbored potential candidate genes that were related to the growth and development of the poplar leaf. Our results enhance the understanding of the molecular mechanism underlying leaf morphological variation in *Populus*. In addition, the multivariate data from a moderate number of regular leaf polar radii could largely represent the leaf shape and exhibited better genomic control in the GWAS of poplar leaf shape.

## Supporting information

**S1 Fig.** Histograms with probability density curves (red) of normal distributions for each univariate trait of L (A), W (B), W31 (C), W21 (D), W32 (E), A (F), and the ratio of L to W (G) in the randomized complete block design derived from the F1 progeny of *Populus deltoides* × *Populus simonii*.
(DOCX)

**S2 Fig.** Manhattan plots of the association analyses without genomic control for each univariate trait of L (A), W (B), W31 (C), W21 (D), W32 (E), A (F), and the ratio of L to W (G) across the 19 chromosomes of the reference genome of *P. trichocarpa*. The horizontal dashed line indicates the genome-wide significant threshold of 5.33, a base 10 logarithm of p-value based on the Bonferroni correction at the 0.05 significant level.
(DOCX)

**S1 Table. The RADseq data information for the 2 parents and 163 progeny in the F1 hybrid population of *Populus deltoides* and *Populus simonii*.**
(DOCX)

**S2 Table. The raw data of leaf length, different widths and area from a randomized complete block design in *Populus*.**
(XLSX)

**S3 Table. Relative difference between leaf measurements with the two software ImageJ and LeafShape.**
(XLSX)

**S4 Table. Correlation coefficients among the leaf traits of L, W, W31, W21, W32, A, and the ratio of L to W in the randomized complete block design derived from the F1 progeny of *Populus deltoides* × *Populus simonii*.**
(DOCX)

**S5 Table. Analysis of variance for the leaf parameters of L, W, W1/3, W1/2, W2/3, and Area in the randomized complete block experiment derived from the F1 progeny of *Populus deltoides* × *Populus simonii*.**
(DOCX)

**S6 Table. Canonical correlation coefficients among the leaf length, widths, area, the length/width ratio, and polar radii in the randomized complete block design derived from the F1 progeny of *Populus deltoides* × *Populus simonii*.**
(DOCX)

**S7 Table. Correlation coefficients between the first principal component of different radius datasets and the leaf length, different widths, or area in the randomized complete block design derived from the F1 progeny of *Populus deltoides* × *Populus simonii*.**
(DOCX)

**S8 Table. Summary of significant SNPs associated to each trait of the four different leaf widths and area without genomic control.**
(DOCX)

**S9 Table. Summary of significant SNPs associated to the ratio of the leaf length to the maximum width without genomic control.**
(DOCX)

**S10 Table. The annotation of candidate genes for the significant SNPs with the non-redundant protein database at NCBI and GO database.**
(XLSX)

**S11 Table. Consistency between significant SNPs for poplar leaf shape identified in the current and previous studies.** The distance of two close SNPs between two different studies is

presented in brackets.
(XLSX)

## Acknowledgments

We thank Professor Huogen Li in Nanjing Forestry University for his great help in establishing the randomized complete block design.

## Author Contributions

**Conceptualization:** Chunfa Tong.

**Data curation:** Dan Yao.

**Formal analysis:** Wenguo Yang, Wei Zhao, Chunfa Tong.

**Investigation:** Wenguo Yang, Dan Yao, Hainan Wu, Wei Zhao, Yuhua Chen.

**Resources:** Chunfa Tong.

**Supervision:** Chunfa Tong.

**Validation:** Hainan Wu, Yuhua Chen.

**Writing – original draft:** Wenguo Yang.

**Writing – review & editing:** Chunfa Tong.

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
