## [Decision Letter · Decision Letter 0]

13 Aug 2021

PONE-D-21-14578

Multivariate Genome-Wide Association Study of Leaf Shape in a Populus deltoides and P. simonii F1 Pedigree

PLOS ONE

Dear Dr. Tong,

Thank you for submitting your manuscript to PLOS ONE. After careful consideration, we feel that it has merit but does not fully meet PLOS ONE’s publication criteria as it currently stands. Therefore, we invite you to submit a revised version of the manuscript that addresses the points raised during the review process.

We look forward to receiving your revised manuscript.

Kind regards,

Karthikeyan Adhimoolam

Academic Editor

PLOS ONE

Journal Requirements:

Reviewers' comments:

Reviewer's Responses to Questions

**Comments to the Author**

1. Is the manuscript technically sound, and do the data support the conclusions?

Reviewer #1: Yes

Reviewer #2: Partly

2. Has the statistical analysis been performed appropriately and rigorously? 

Reviewer #1: Yes

Reviewer #2: Yes

3. Have the authors made all data underlying the findings in their manuscript fully available?

Reviewer #1: Yes

Reviewer #2: Yes

4. Is the manuscript presented in an intelligible fashion and written in standard English?

Reviewer #1: Yes

Reviewer #2: Yes

5. Review Comments to the Author

Reviewer #1: Overall the manuscript is clear and well written and the results support the conclusions. It presents an interesting analysis of leaf shape.

I have several specific points that need clarification.

1. The abstract is clear and the results are well presented but it doesn't say why poplar or why leaf shape. A sentance explaining this in the abstract would make it clearer because at present the reason for using poplar isn't really standing out.

2. Line 54 in the introduction- polygenic is a standard term and doesn't really need defining here

3. The QTL analysis needs more detail in the methods section.

4. Line 91 needs another look at the grammar

5. I know this is mentioned in the introduction but in the methods you could make it clear which is the male and female parent in your cross.

6. Line 153 a rather than an SNP

7. Line 186 it isn't clear where the 2,244 samples came from. The methods state 3 blocks and 6 leaves per block

8. Line 239 The sentance needs rewriting, should it be -with few SNPs on chromosomes

Reviewer #2: This manuscript described QTL mapping in poplar. A segregating mapping population was used as plant material, while a GWAS model was employed for data analysis. The result is interesting; however, I do have some concerns. The author should consider the below points.

1. QTL mapping was conducted with segregating population. Did the author considered the population structure (Q) and kinship (K)? How did the mvLMM work? I noticed the model Y= XB+ZG + E, it seems there is similar parameters of Q and K in this model, if so, how did the author calculate them?

2. The author investigated 100 genes for a significant SNPs, why 100? Usually, the physical distance would be used as a threshold.

3. The QTL results seemed not good. The author compared their results with some previous reports. But I found they missed a very similar study. In that study, P. deltoides and P. simonii were also used as parents for a F1 population. See https://doi.org/10.1007/s00425-018-2958-y, is there any overlapping regions between the 2 studies?

6. PLOS authors have the option to publish the peer review history of their article (what does this mean?). If published, this will include your full peer review and any attached files.

Reviewer #1: No

Reviewer #2: No

---

## [Author Response · Author response to Decision Letter 0]

19 Aug 2021

Response to Reviewers’ Comments:

Reviewer #1: Overall the manuscript is clear and well written and the results support the conclusions. It presents an interesting analysis of leaf shape.

I have several specific points that need clarification.

1. The abstract is clear and the results are well presented but it doesn't say why poplar or why leaf shape. A sentance explaining this in the abstract would make it clearer because at present the reason for using poplar isn't really standing out.

RE: Thank you very much for your suggestion. We added a sentence in Abstract to address the importance of poplar and its leaf shape as follows: “Most poplar species are of great economic and ecological values and their leaf morphology can be a predictor for wood productivety and environment adaptation”.

2. Line 54 in the introduction- polygenic is a standard term and doesn't really need defining here

RE: Thanks. We deleted the words in the bracket as you suggested.

3. The QTL analysis needs more detail in the methods section.

RE: Thank you for this suggestion. After Line 140, we described the mvLMM for details in a single data form as follows:

“ 

(1)

where is the lth polar radius of the kth tree leaf of the jth clone in the ith block; is the overall mean of the lth polar radius; is the effect of the ith block; is the genotype effect of the jth clone at any tested SNP; is the polygenic background effect of the jth clone; and is the residual effect. It is assumed that and are fixed effects, while and are random effects with , and . In matrix form, model (1) can be written as …”.

4. Line 91 needs another look at the grammar:

RE: Thanks. We modified the sentence as “we identified many more SNPs significantly associated with leaf shape than those detected in previous studies”.

5. I know this is mentioned in the introduction but in the methods you could make it clear which is the male and female parent in your cross.

RE: Thank you very much for your suggestion. In Line 103, we changed “P. deltoides and P. simonii” to “the female P. deltoides and the male P. simonii”.

6. Line 153 a rather than an SNP

RE: Thanks. We modified “an SNP” as “any single SNP”.

7. Line 186 it isn't clear where the 2,244 samples came from. The methods state 3 blocks and 6 leaves per block

RE: Thanks for your comments. We inserted a sentence there to explain that there were a few missed samples in some plots as follows: “Some plots had missed samples due to the damage from pest, disease, poor rooting ability, or other unknown reasons”.

8. Line 239 The sentance needs rewriting, should it be -with few SNPs on chromosomes

RE: Thanks for pointing out this grammar error. We modified it as “with a few SNPs on chromosomes …”.

Reviewer #2: This manuscript described QTL mapping in poplar. A segregating mapping population was used as plant material, while a GWAS model was employed for data analysis. The result is interesting; however, I do have some concerns. The author should consider the below points.

1. QTL mapping was conducted with segregating population. Did the author considered the population structure (Q) and kinship (K)? How did the mvLMM work? I noticed the model Y= XB+ZG + E, it seems there is similar parameters of Q and K in this model, if so, how did the author calculate them?

RE: Thank you very much for your comments. 

1) Because our samples were from a single full-sib family, the population was so simple that it is improper to apply the population structure (Q) model to the current GWAS. In fact, the mvLMM involved in the kinship matrix (K) with the relationship of A=2K, where A is the additive relationship matrix and the K was estimated from the genetic theory [38]. The relationship of A and K was clearly described in Lines 150-152. 

2) Y= XB+ZG + E is the mvLMM model in matrix form, which is very complicated. However, in theory, it is more mature because it belongs to linear model and is widely applied. The F statistics was presented as Equation (3) for testing a single SNP, which can be calculated with the R package EMMREML (https://cran.r-project.org/web/packages/EMMREML). Please see Lines 171-174 for details.

2. The author investigated 100 genes for a significant SNPs, why 100? Usually, the physical distance would be used as a threshold.

RE: Thanks. We took the so-called proximate strategy to investigate the nearby genes of each significant SNP (Monclus et al. 2012; Geng et al. 2015; Su et al. 2017; Vanous et al. 2018). Those genes that have annotations related to the leaf growth and development were chosen as candidate genes, which were listed in Table 3. However, in literatures, there is no conclusion about how many nearby genes should be provided for possible candidates. We chose 100 nearby genes, expecting to find as many as possible candidate genes for a significant SNP. In order to clarify this point, we cited the 4 literatures in Line 177. 

Similarly, the threshold of a physical distance could be used to investigate the nearby genes, but the number of candidate genes certainly varied along QTLs. 

Monclus et al. 2012. Integrating genome annotation and QTL position to identify candidate genes for productivity, architecture and water-use efficiency in Populus spp. BMC Plant Biology 12:173 DOI 10.1186/1471-2229-12-173.

Geng et al. 2015. A genome-wide association study in catfish reveals the presence of functional hubs of related genes within QTLs for columnaris disease resistance. BMC Genomics 16:196 DOI 10.1186/s12864-015-1409-4.

Su et al. 2017. High density linkage map construction and mapping of yield trait QTLs in maize (Zea mays) using the genotyping-by-sequencing (GBS) technology. Frontiers In Plant Science 8 DOI 10.3389/fpls.2017.00706.

Vanous et al. 2018. Association mapping of flowering and height traits in germplasm enhancement of maize doubled haploid (GEM-DH) lines. Plant Genome 11(2):1-14 DOI 10.3835/plantgenome2017.09.0083.

3. The QTL results seemed not good. The author compared their results with some previous reports. But I found they missed a very similar study. In that study, P. deltoides and P. simonii were also used as parents for a F1 population. See https://doi.org/10.1007/s00425-018-2958-y, is there any overlapping regions between the 2 studies?

RE: Thanks for your comments. Yes, this literature was about QTL study for leaf shape, but it was based on linkage maps and no physical position information was available for the QTLs detected. In Lines 338-339, we wrote “S11 Table lists those significant SNPs associated with leaf shape in the current study and in three recent studies [17, 18, 26], excluding the previous QTL studies because no position information was available on the physical maps for those QTLs related to poplar leaf shape [11, 25, 27, 28]”. Nevertheless, we cited this literature there as reference 63.

---

## [Decision Letter · Decision Letter 1]

7 Sep 2021

PONE-D-21-14578R1

Multivariate Genome-Wide Association Study of Leaf Shape in a Populus deltoides and P. simonii F1 Pedigree

PLOS ONE

Dear Dr. Chunfa Tong,

Thank you for submitting your manuscript to PLOS ONE. After careful consideration, we feel that it has merit but does not fully meet PLOS ONE’s publication criteria as it currently stands. Therefore, we invite you to submit a revised version of the manuscript that addresses the points raised during the review process.

ACADEMIC EDITOR:  Reviewers raised concerns and were not satisfied with the response from the authors. Therefore, I suggest authors writing a detailed response.  Recommended for major revision.

We look forward to receiving your revised manuscript.

Kind regards,

Karthikeyan Adhimoolam

Academic Editor

PLOS ONE

Journal Requirements:

Additional Editor Comments (if provided):

Well, reviewers raised concerns and were not satisfied with the response from the authors. Thus, I suggest authors writing a detailed response. Recommended for major revision.

Reviewers' comments:

Reviewer's Responses to Questions

**Comments to the Author**

1. If the authors have adequately addressed your comments raised in a previous round of review and you feel that this manuscript is now acceptable for publication, you may indicate that here to bypass the “Comments to the Author” section, enter your conflict of interest statement in the “Confidential to Editor” section, and submit your "Accept" recommendation.

Reviewer #1: All comments have been addressed

Reviewer #2: (No Response)

2. Is the manuscript technically sound, and do the data support the conclusions?

Reviewer #1: Yes

Reviewer #2: (No Response)

3. Has the statistical analysis been performed appropriately and rigorously? 

Reviewer #1: Yes

Reviewer #2: (No Response)

4. Have the authors made all data underlying the findings in their manuscript fully available?

Reviewer #1: Yes

Reviewer #2: (No Response)

5. Is the manuscript presented in an intelligible fashion and written in standard English?

Reviewer #1: Yes

Reviewer #2: (No Response)

6. Review Comments to the Author

Reviewer #1: All of the comments and suggestions that I made have now been addressed and clearly answered. The manuscript is now much improved after all comments and suggestions have been taken on board.

Reviewer #2: In the last round of review, I asked 3 questions, but the author did not provide the answerers directly. The first question is about the calculation of parameters in the association model. Yes, the equation was provided in the M&M, however, how did you match your data to each parameter. For example, in the GWAS model, it is not right to use all SNPs for Q calculation, but all SNPs should be used for K calculation. So, how did you calculate A and K?

The second question is for determination of candidate genes. Frankly, I don`t think 100 genes around the significant SNP were a right selection. In different regions of genome, the density of genes was different. In linkage-based QTL mapping, 95% confident interval was used, while LD length was used in GWAS. So, I think there should be also a strategy for you to define the candidate QTL region in the model you used. Yes, different strategies, including 100 genes, were used in some published paper, but this doesn`t mean all of them were reasonable.

The third question is for comparison between your results with a previous very similar study. The author didn`t perform the comparison because they could not find physical positions of SNP in that paper. Is that true? I am very sure there was physical position for each SNP. Also, the authors also provided physical position for each candidate region in that study.

I found the quality of the study is far above the quality of writing in this paper. Frankly, your work could be published some other high-reputational journals. The author should carefully revise their manuscript.

7. PLOS authors have the option to publish the peer review history of their article (what does this mean?). If published, this will include your full peer review and any attached files.

Reviewer #1: No

Reviewer #2: No

---

## [Author Response · Author response to Decision Letter 1]

5 Oct 2021

Response to Reviewers’ Comments:

Reviewer #1: All of the comments and suggestions that I made have now been addressed and clearly answered. The manuscript is now much improved after all comments and suggestions have been taken on board.

RE: Thank you very much for your positive comments.

Reviewer #2: In the last round of review, I asked 3 questions, but the author did not provide the answerers directly. The first question is about the calculation of parameters in the association model. Yes, the equation was provided in the M&M, however, how did you match your data to each parameter. For example, in the GWAS model, it is not right to use all SNPs for Q calculation, but all SNPs should be used for K calculation. So, how did you calculate A and K?

RE: Thank you very much for pointing out this issue again. 

1) In this study, the clones were from a full-sib family. In theory, for any two clones, the coefficient of kinship (constituted to K matrix) is expected to be 0.25 (Loiselle et al. 1995; Lynch and Walsh 1998). This led to the relationship matrix (A=2K) with elements of ones on the diagonal and 0.5 elsewhere (Lynch and Walsh 1998; Bae et al. 2016). Here, it is unnecessary to calculate (in fact, to estimate) the matrix K with SNP data because the pedigree is fully known and each value in the K matrix is equal to 0.25 or 0.5. We think we have described this issue clearly and cited related literatures in Lines 152-159.

Certainly, in most GWAS studies where the natural population were used and the pedigrees were usually unknown, it need to estimate the kinship matrix with SNP data. However, from a statistical point of view, theoretical values are usually better than estimators because the accuracy of estimators depends not only on the completeness of sampling data but also on the estimate method.

2) Population structure (Q) was originally used in human GWAS for overcoming genomic inflation because the population was considered natural and more complicated. However, the Q method is not a “golden rule” for all GWAS, and sometimes, it failed to control genomic inflation (Kim 2019; https://doi.org/10.1101/647768). In the current study, we established the multivariate linear mixed model for GWAS of leaf shape according to the traditional randomized complete block design. Through adjusting the dimension of the regular polar radii data, we found that a moderate dimensional data can be used to successfully control genomic inflation and obtain a better result in finding significant SNPs associated to leaf shape. We think that our mvLMM for GWAS was successfully established and every parameter was described clearly. Moreover, the whole calculation for parameter estimates and significant hypothesis test was implemented with the R package EMMREML (https://cran.r-project.org/web/packages/EMMREML).

Bae, H., S. Monti, M. Montano, M.H. Steinberg, T.T. Perls et al., 2016 Learning Bayesian networks from correlated data. Sci Rep 6: 25156. https://doi.org/10.1038/srep25156

Loiselle, B.A., V.L. Sork, J. Nason, and C. Graham, 1995 Spatial genetic structure of a tropical understory shrub, Psychotria officinalis (Rubiaceae). American Journal of Botany 82: 1420-1425. https://doi.org/10.1002/j.1537-2197.1995.tb12679.x

Lynch, M., and B. Walsh, 1998 Genetics and Analysis of Quantitative Traits. Sunderland, MA, USA: Sinauer Associates, Inc.

The second question is for determination of candidate genes. Frankly, I don`t think 100 genes around the significant SNP were a right selection. In different regions of genome, the density of genes was different. In linkage-based QTL mapping, 95% confident interval was used, while LD length was used in GWAS. So, I think there should be also a strategy for you to define the candidate QTL region in the model you used. Yes, different strategies, including 100 genes, were used in some published paper, but this doesn`t mean all of them were reasonable.

RE: Thank you for pointing out this issue again. As you suggested, we modified the investigation of candidate genes with the method of LD analysis. But, we found the average LD length was estimated to be ~650 bp in our previous study (Chen et al., 2021; https://www.doi.org/10.1093/g3journal/jkaa053). It is so short and cannot be properly used as a range for candidate genes. Therefore, we considered genes within a LD block that contained a significant SNP for investigating candidate gene. The corresponding sections in M&M and Results as well as Table 3 and S10 Table were thoroughly modified.

The third question is for comparison between your results with a previous very similar study. The author didn`t perform the comparison because they could not find physical positions of SNP in that paper. Is that true? I am very sure there was physical position for each SNP. Also, the authors also provided physical position for each candidate region in that study.

RE: Thank you for pointing out this issue again and sorry for not carefully considering it in the previous manuscript. In fact, the accurate QTL position was described in genetic distance (cM), but not available in physical distance (bp). However, their physical position can be determined by flanking SNPs. Therefore, we performed the comparison by adding the work as you suggested. Please see the result in in S11 Table. Accordingly, the last paragraph in Discussion was modified.

I found the quality of the study is far above the quality of writing in this paper. Frankly, your work could be published some other high-reputational journals. The author should carefully revise their manuscript.

RE: Thank you very much for this positive comment. We carefully revised those paragraphs for investigating candidate genes and the comparison study with previous similar works.

---

## [Decision Letter · Decision Letter 2]

18 Oct 2021

Multivariate Genome-Wide Association Study of Leaf Shape in a Populus deltoides and P. simonii F1 Pedigree

PONE-D-21-14578R2

Dear Dr. Chunfa Tong,

We’re pleased to inform you that your manuscript has been judged scientifically suitable for publication and will be formally accepted for publication once it meets all outstanding technical requirements.

Kind regards,

Karthikeyan Adhimoolam

Academic Editor

PLOS ONE

Additional Editor Comments (optional):

Reviewers' comments:

Reviewer's Responses to Questions

**Comments to the Author**

1. If the authors have adequately addressed your comments raised in a previous round of review and you feel that this manuscript is now acceptable for publication, you may indicate that here to bypass the “Comments to the Author” section, enter your conflict of interest statement in the “Confidential to Editor” section, and submit your "Accept" recommendation.

Reviewer #1: All comments have been addressed

Reviewer #2: All comments have been addressed

2. Is the manuscript technically sound, and do the data support the conclusions?

Reviewer #1: Yes

Reviewer #2: Yes

3. Has the statistical analysis been performed appropriately and rigorously? 

Reviewer #1: Yes

Reviewer #2: Yes

4. Have the authors made all data underlying the findings in their manuscript fully available?

Reviewer #1: Yes

Reviewer #2: Yes

5. Is the manuscript presented in an intelligible fashion and written in standard English?

Reviewer #1: Yes

Reviewer #2: Yes

6. Review Comments to the Author

Reviewer #1: (No Response)

Reviewer #2: (No Response)

7. PLOS authors have the option to publish the peer review history of their article (what does this mean?). If published, this will include your full peer review and any attached files.

Reviewer #1: No

Reviewer #2: No

---

## [Editor Report · Acceptance letter]

20 Oct 2021

PONE-D-21-14578R2 

Multivariate Genome-Wide Association Study of Leaf Shape in a *Populus deltoides* and *P. simonii* F_1_ Pedigree 

Dear Dr. Tong:

I'm pleased to inform you that your manuscript has been deemed suitable for publication in PLOS ONE. Congratulations! Your manuscript is now with our production department. 

Kind regards, 

on behalf of

Dr. Karthikeyan Adhimoolam 

Academic Editor

PLOS ONE